# Challenges and Opportunities in the Genetic Analysis of Inherited Retinal Dystrophies in Africa, a Literature Review

**DOI:** 10.3390/jpm13020239

**Published:** 2023-01-29

**Authors:** Oscar Onyango, Marianne Mureithi, Dennis Kithinji, Walter Jaoko, Kaoru Fujinami

**Affiliations:** 1Department of Ophthalmology, Kenyatta National Hospital, Nairobi P.O. Box 20723-00202, Kenya; 2KAVI-Institute for Clinical Research, University of Nairobi, Nairobi 00202, Kenya; 3Medright Consulting Ltd., Maua P.O. Box 254715149694-60600, Kenya; 4Laboratory of Visual Physiology, Division of Vision Research, National Institute of Sensory Organs, National Hospital Organization Tokyo Medical Center, Tokyo 152-8902, Japan

**Keywords:** inherited, retinal, dystrophy, indigenous, black, Africa, genetics

## Abstract

Inherited retinal dystrophies (IRDs) are a global problem that is largely unaddressed, especially in Africa. Black indigenous Africans are rarely represented in research that develops genetic tests and genetic therapies for IRDs, yet their genomes are more diverse. The aim of this literature review is to synthesize information on the IRD genetic research conducted among indigenous black Africans to identify challenges and opportunities for progress. PubMed was searched to identify empirical publications reporting the genetic analysis of IRDs among indigenous Africans. A total of 11 articles were selected for the review. Based on the information in the articles, the main genetic testing methods in use include next-generation, whole exome, and Sanger sequencing. The main IRDs characterized by the genetic tests include retinitis pigmentosa, Leber Congenital Amaurosis, Stagardt disease, and cone dystrophy. Examples of implicated genes include *MERTK*,* GUCY2D*,* ABCA4*, and *KCNV2* for the four IRDs, respectively. Research activities on the genetics of IRDs are generally scanty in Africa. Even in South Africa and North Africa where some research activities were noted, only a few indigenous black Africans were included in the study cohorts. There is an urgent need for genetic research on IRDs, especially in East, Central, and West Africa.

## 1. Introduction

Inherited retinal dystrophies (IRDs) comprise progressive genetic diseases of the eye, whose patients present with loss of visual field, night vision, central acuity, and/or color [1]. Retinitis pigmentosa, the most common form of IRD, is estimated to affect 1 in every 3500 people globally [2]. Two-thirds of blindness cases in Cape Town, South Africa, are attributable to posterior segment diseases that include some inherited retinal dystrophies, diabetic retinopathy, and glaucoma [3]. As the prevalence of blindness in Cape Town is 1.4% [3], about 32,000 people in the city are affected by posterior segment diseases. It is estimated that about 11,600 indigenous Africans in South Africa are affected by IRDs [3]. Therefore, IRDs could be responsible for a substantial proportion of the blindness cases in Cape Town, South Africa, and the rest of Africa. 

The Division of Human Genetics at the University of Cape Town has been the center of research work on IRDs in sub-Saharan Africa for the past 30 years [4], but it only focuses on IRDs in South Africa [3]. Only 19% of the families in the registry of IRD research in South Africa have been genetically diagnosed [3]. The scarcity of empirical information regarding the IRDs in Africa and particularly the rest of sub-Saharan Africa calls for increased research activity on the issue beyond South Africa.

IRDs occur at any point in life from birth to advanced age [1]. About 9% of blindness cases globally are due to age-related macular degeneration [5]. Approximately 288 million people globally are projected to have age-related macular degeneration by 2040 [5,6]. The global prevalence of retinitis pigmentosa (RP), an IRD, is estimated to be between 1 in 8357 to 1 in 3454 [2,7]. The prevalence of Leber Congenital Amaurosis (LCA) 2, which is an IRD that shares some genetic mutations with RP, is between 1 in 81,000 and 1 in 33,000 [7].

Over 270 genes have been identified as responsible for retinal dystrophies [8]. The rate of identification of novel variants of IRDs dramatically increases when people with African ancestry are included in IRD studies [3]. The genotyping-based microarrays that test for IRD-associated mutations are only scantly useful among Africans because they were developed based on data from predominantly European populations [9], whose genetics are not as diverse as Africans’ [3]. Whole exome sequencing and other genetic studies among indigenous Africans with IRDs are revealing unique mutations associated with the various IRDs [3,10].

The mainstay of IRD testing is an electroretinogram, but it only provides a phenotypic diagnosis [11]. It measures the photoreceptor function through the a-wave and the inner retinal-bipolar cell function via the b-wave [1]. Fundus autofluorescence is used to show the retinal pigment epithelium (RPE) photoreceptor activity, which can be the basis for the definitive diagnosis of conditions such as Stargardt disease [1,12]. Optical coherence tomography (OCT) entails the application of imaging analogous to ultrasound to facilitate the visualization of the retina in detail, which allows ophthalmologists to measure retinal thickness, segment retinal layers, and assess the progress of disease [1].

The emerging trend in the diagnosis of IRDs is the use of modern genomic techniques to identify diseases and understand their mechanisms [1,11]. Genetic testing entails using next-generation sequencing or whole exome panels, among other molecular techniques, to identify IRDs based on their associated mutations [13]. It should be conducted early in the diagnostic process because it can inform the selection of gene therapy [1]. Africa needs genome-based eye care programs to promote personalized treatment because IRDs have ethnic patterns [5]. Africans are underrepresented in previous genetic research on IRDs, yet their genetic variations make them valuable for discovering novel genes and variants in patients with IRDs [3].

The current review aims to synthesize the findings of empirical studies that have been conducted through genetic testing to characterize inherited retinal dystrophies in Africa. The researchers will assess the extent to which studies have explored the vast genomic diversity among Africans to advance the molecular diagnosis of IRDs among Africans [3]. The review will show the state of evidence regarding progress toward the genetic characterization of Africans with IRDs. It will reveal knowledge gaps that ophthalmology researchers can focus on filling through exploratory and experimental research on the genetics of IRDs. 

## 2. Materials and Methods

MEDLINE database was searched through the PubMed search engine to identify relevant articles. The search process entailed using Boolean operators to group search terms for specific, yet thorough identification of relevant articles. The search terms included “inherited” OR “hereditary” OR “heritable” OR “genetic” AND “retinal dystrophy” OR “retinal disease” OR “retinopathy” OR “retinitis pigmentosa” AND “Africa” OR “Africans” OR “Sub-Saharan Africa” OR “SSA.” The time of publication was not restricted considering that the research topic is scantly covered in Africa. 

Only original research articles, abstracts, and conference proceedings that addressed the diagnosis of IRDs among Africans using genetic analysis were considered for selection. Only articles published in English were chosen. Articles that did not focus on advances in the diagnosis of IRDs were excluded. Non-human studies were also excluded. 

The PubMed search process yielded 21 records. Two additional articles were identified by perusing the reference lists of the relevant articles. The records were screened by checking their titles to remove duplicates—20 records remained. The abstracts of the records were reviewed to determine the relevance of the publications in answering the research question. Thus, 13 articles were identified as relevant as they met the inclusion and exclusion criteria. Upon reading the full-text articles, two more articles were excluded for inappropriateness in answering the research question. Therefore, 11 articles were included in this review (Figure 1).

Information from the included publications (Table 1) was synthesized thematically. The focus was on the contributions of the research in advancing the diagnosis and precision medicine for IRDs. African regions where the studies were conducted were noted in the information synthesis to facilitate the identification of countries that have contributed to the advancements and identify opportunities that researchers can leverage to supplement the current efforts. 

## 3. Results

### 3.1. Summary of Selected Articles

Bouzidi et al. [14] compiled published data including molecular diagnoses from 413 families with IRDs or inherited optical neuropathies (ION). Index cases comprised 82.8% IRDs and 17.8% ION. Forty-six of the families had a North African origin. Notably, 28 of the families were Tunisian. The majority of the North African families (93.5%) had homozygous variants; 34 variants in nine genes. The authors characterized many founder mutations in small endogamous communities.

Maltese et al. [15] conducted genetic testing using multiple approaches to diagnose IRD cases conclusively. Although their study was conducted in three Italian hospitals, 26.8% (33/123) of the included patients were North Africans (Egypt, 11; Tunisia, 7; Morocco, 7; Cape Verde, 3; Senegal, 2; Mauritania, 1; and Nigeria, 1). The patients had been referred to the IRD units of the hospitals for genetic diagnosis and treatment. Genetic tests of two-thirds (21/33) of African patients turned out to be positive for IRDs, as shown in Figure 2 [15].

Bouzidi et al. [16] conducted clinical and genetic studies among three families in Morocco with retinitis pigmentosa (RP) phenotypes. Through whole exome sequencing (WES), they identified mutations in individuals with RP, expanding the genetic diagnosis of RP. The three families were unrelated. Two of the families were consanguineous [16]. 

Greenberg et al. [17] analyzed the register at the DNA banking center for RP at the Department of Human Genetics, University of Cape Town to characterize RP in South Africa. By 1993, when the article was published, the department had been documenting families with RP for the previous 21 years [17]. The researchers report the frequencies of various types of RP based on the mode of inheritance as determined from genetic analysis of samples collected from 75 families with RP in South Africa. However, they did not identify the affected genes and implicated mutations.

Perrault et al. [18] analyzed the genetic mutations in eight multiplex families that had been characterized as having pathogenic genes associated with LCA in chromosome 17p13.3. Seven of the eight families had North African ancestry. Six of the eight families were consanguineous [18].

Maggi et al. [19] investigated unresolved indigenous African IRD cases by analyzing samples of patients with a family history suggestive of autosomal recessive RP, autosomal dominant RP, or X-linked RP (17 patients). They also analyzed samples from 15 probands with sporadic or recessive macular degeneration deposited at the University of Cape Town IRD registry [19]. They used next-generation PCR and sequencing to overcome the limitations of WES to sequence long-range PCR amplicons of DNA samples from an IRD cohort of African ancestry to study the *RPGR* exon ORF15 variants [19]. 

McKie et al. [20] conducted a multinational study that covered South Africa to enhance the understanding of the genetics of RP, particularly the RP13 locus for dominant RP on chromosome 17p13.3. They applied PCR, segregation analysis, reverse transcriptase PCR, and Northern blot analysis to characterize the RP13 interval [20]. They used the NIX nucleotide amplification package to predict the implicated genes. 

Camuzat et al. [21] mapped *LCA1*, a gene implicated in LCA, to chromosome 17p13 and demonstrated genetic heterogeneity of LCA. Out of the 18 families included in their study, 7 were from North Africa. Among the seven consanguineous families in the study, five were North African.

Falfoul et al. [22] reported the genetic features of a cohort of patients with enhanced-S-cone syndrome (ESCS) in Tunisia. The cohort comprised six consanguineous families. Out of the nine patients, seven could be characterized using genetic analysis. The analysis revealed *NR2E3* mutations in five patients. The other two patients had NR2E3 + RHO digenism; they were relatives [22].

Roberts et al. [23] utilized the finding that *ABCA4* mutations are implicated in the occurrence of retinopathies in South Africa to develop a test for screening of *ABCA4*-associated retinopathies (AAR). The test was designed to detect seven *ABCA4* mutations commonly implicated in IRDs including p.Arg152, p.Arg602Trp, c.768G > T, p.Cys1490Tyr, p.Gly863Ala, p.Leu2027Phe, and c.5461–10T > C. However, only 17 of the 181 samples from affected probands with AAR were from indigenous Black Africans. The majority (152/181) were Caucasians, as shown in Figure 3.

September et al. [24] investigated the mutation spectrum implicated in Stargardt disease in South Africa. The sample comprised 64 probands with Stargardt disease phenotype. Although the study was conducted in South Africa, the probands were Caucasians. They identified 57 *ABCA4* disease-associated alleles, with the most common variants being *C1490Y*,* R152X*,* L2027F*,* V256splice*,* R602W*, and *2588G-C* mutations. 

### 3.2. Testing Methods Used 

High throughput next-generation sequencing (NGS) is the main method used to characterize genetic mutations in the diagnosis of IRDs [14]. Specific sets of genes are also targeted using gene sequencing panels for genetic testing. Precise identification of genotypes is sometimes obtained using Sanger sequencing and multiplex ligation-dependent probe amplification (MLPA). Whole exome sequencing is also applied in the genetic testing of IRDs [16]. Gene-specific chips such as the ABCR400 chip and mutation-targeting rapid genetic tests such as the ones targeting ABCA4 mutations are sometimes used [23]. Other methods that have been previously useful in genetic testing include sequence derivation via P1-derived artificial chromosome (PAC) analysis, vectorette PCR, fragment detection in Southern blotting, single-strand conformational polymorphism analysis and heteroduplex analysis, and dye terminator chemistry cycle sequencing [20,21].

A genetic test for an IRD is considered positive if a pathogenic or a likely pathogenic variant is found in dominant and X-linked genes, with or without family history [15]. The presence of variants of unknown significance (VUS) on the dominant and x-linked genes has to be supported by results of a family segregation study or family history to be the basis of declaring a genetic test positive. Variants in recessive genes have to be at least two in number and supported by confirmation of biallelism in a family segregation study for a genetic test to be declared positive [15].

### 3.3. Common Genetically Diagnosed Diseases 

#### 3.3.1. Retinitis Pigmentosa

RP diagnosis was made in 18.4% (76/413) of the families in the study by Bouzidi et al. [14]. RP was more prevalent among African patients than among patients from other regions referred to the Italian hospitals where the study was conducted [15]. The proportion of African patients with RP among those that genetically tested positive was 62% (13/21) [15]. 

About 20 genes were identified in 45 non-Jewish families with RP in North Africa [14]. Five of the 20 genes were responsible for 53.3% of the observed RP phenotypes. The most frequently mutated genes include *MERTK* (8/45 families), *PDE6B* (5/45 families), *CERKL* (4/45 families), and *RP1* (4/45 families) [14]. The genes identified as responsible for RP among North Africans in the study by Maltese et al. [15] include *MERTK*,* CNGA1*,* CNGB1*,* PDE6A*,* PDE6B*,* NR2E3*,* CERKL*,* ABCA4*, and *RDH12*. Bouzidi et al. [16] identified pathogenic mutations in *ABCA4* and *CRB1* genes in RP patients. Additionally, a *PDE6B* mutation, which is associated with cystoid macular edema, was detected in one patient [16]. McKie et al. [20] identified *RPA1*,* PEDF*,* PLI*,* SKIP*,* PITPn-a*,* MY01-β*,* KIAA00149*, and *IMPC23B* as the genes implicated in the RP. They also showed that *PRPC8* is implicated in the RP13 form of RP [20].

The mutations identified in the three Moroccan families through WES include c.1690G > T and c.1913C > T in the *CRB1* gene, c.1920+2T > C in the *PDEGB* gene, and c.5908C > T and c.6148G > C in the *ABCA4* gene [16]. In the study by Maggi [19], two novel pathogenic frameshift mutations, namely c.2470_2471delGG and c.2457_2460delAGAG, were identified, with the former mutation being recurrent in two families, while the latter was identified in four families. Another recurrent mutation detected in the study is c.2964_2965delGG [19]. All of the families were of different ethnolinguistic groups.

#### 3.3.2. Leber Congenital Amaurosis (LCA)

Camuzat et al. [21] confirmed that the *LCA1* gene is located on chromosome 17p13 and identified its loci as between D17S938 and D17S1353. Hence, they enhanced the process of identifying the genes responsible for the LCA. Out of the 41 families that were diagnosed with LCA, 36 had the homozygous variant, while 5 had the heterozygous compound variants [21]. In the study by Maltese et al. [15], one patient had LCA. Twelve genes were affected. *GUCY2D* mutations were the most common as they were present in 46.3% of the families. The other main gene mutations include *LCA5* (4 families), *RPGRIP1* (3 families), *RD3* (3 families), and *NMNATI* (3 families). The *RPE65* gene was implicated in the patient diagnosed with LCA in the research by Maltese et al. [15]. Perrault et al. [18] identified *LCA1* mutations in 8/15 LCA families in their study. They also reported two frameshift mutations (nt 460 del C, nt 693 del C) and two missense mutations (F589S) in the *GUC2D* gene that codes for the retinal guanylate cyclase in four unrelated North Africans [18]. 

#### 3.3.3. Stargardt Disease (STGD)

STGD4 was diagnosed in eight Tunisian families [14]. In the study by Maltese et al. [15], 2/33 patients were diagnosed with STGD. Homozygous *ABCA4* variants characterized the diagnoses among the Tunisian families [14]. The most common mutations were p.Arg681Ter (3 families) and p.Glu1087Lys (2 families) [14]. Roberts [23] successfully used tests developed based on common *ABCA4* mutations to diagnose Stargardt macular dystrophy, which links the disease to the mutations. In the study by September et al. [24], 10 *ABCA4* disease-associated haplotypes were detected. Two of the haplotypes had the most common disease-associated variant, *C1490Y*, which was present in 19/64 subjects and absent in 392 control chromosomes. 

#### 3.3.4. Cone Dystrophy 

In the study by Maltese et al. [15], only one out of 33 patients was diagnosed with cone dystrophy. Genetic studies identified 18 families with the causal genes for cone dystrophy [14]. There were 15 and 2 homozygous and heterozygous compound mutations, respectively, in 9 different genes [14]. Half of the 18 families had mutations in the *ABCA4* and *KCNV2* genes [14].

#### 3.3.5. Syndromic IRDs 

Bouzidi et al. [14] genetically characterized 24 syndromic IRDs among North Africans. The main syndromes were Usher (41.2%) and Bardet Biedl (31.1%). Sixty-one families in North Africa have been genetically characterized as having Usher syndrome. The most commonly mutated genes include *MYO7A* (34 families), *USH2A* (8 families), *CDH23* (6 families), and *USH1G* (4 families) [14]. Similarly, Bardet–Biedl syndrome was reported in 46 families, 28 of whom were Tunisian. Mutations in nine genes were observed. The most common ones include *BBS1* (10 families), *BBS2* (8 families), *BBS5* (8 families), *BBS8* (6 families), and *BBS4* (5 families) [14]. Other diagnoses detected among Africans include albinism (*GPR143* and *TYR* genes), Senior–Loken Syndrome (*NPHP4* gene), and X-linked retinoschisis (*RS1* gene) [15].

### 3.4. Diagnostic Challenges 

IRDs are underdiagnosed in Africa. For example, the Usher syndrome diagnosis could be underestimated in North Africa because of a lack of systematic access to ophthalmological examinations [14]. Most IRD families are located in remote areas, where accessing an ophthalmologist is difficult [14]. Out of the 75 RP families that underwent DNA analysis in 1993 for genetic diagnosis, only six RP kindreds were black [17]. The rest were Caucasian (47), mixed ancestry (20), and Indian/Asian (2) [17]. In the study by Roberts et al. [23] to assess a test for the diagnosis of *ABCA-4* associated retinitis (AAR), only 9.4% of patients (17/181) were indigenous Black Africans. Mainly non-indigenous Africans underwent the ABCR400 microarray screening, yet the study was performed in South Africa. Among the 17 indigenous Africans, only 1 was in the cohort of 72 unrelated patients who were screened with the locally-developed diagnostic assay targeting seven *ABCA4* mutations. Similarly, in a study by September et al. [24] conducted in South Africa, the patients were mainly Caucasians (59/64); only 3/64 were indigenous Black Africans.

Poor access to diagnostic facilities is an issue of concern in Africa. Genetic testing is not considered a health priority in most African countries [15]. The high throughput next-generation sequencing required to genetically diagnose IRDs is difficult to access locally [14]. The researchers that analyzed three families with RP in Morocco had to outsource WES from BGI Tech Solutions in China [16].

It may be challenging to accurately diagnose IRDs genetically in some cases. Some mutations such as the *RPGR* exon ORF15 variants have low sequence complexity or may have indels. Thus, sequence data analysis may be difficult when handling them [19]. Whole exome sequencing is not appropriate to detect such mutations as it may not adequately cover the region. Standard NGS analysis tools may generate miscalling of variants. Secondary analysis pipelines such as preceding NGS with long-range PCR amplification can be helpful. Maggi et al. [19] identified three pathogenic variants that have been previously reported and two novel pathogenic variants that had not been characterized in 11 patients. Thus, the secondary analysis pipeline can be leveraged to avoid false positive and negative variant calls that would have occurred if conventional NGS was applied.

### 3.5. Diagnostic Opportunities 

Consanguinity in some African families is advantageous in the genetic study of IRDs because of the high rate of recessive expression [15]. The common ancestry creates opportunities for the identification of more prevalent genes. The genetic tests also provide empirical evidence of the influence of consanguinity in the occurrence of IRDs. Consanguinity was established in 71.4% of the 413 families in the study by Bouzidi et al. [14], hence the predominance of the autosomal recessive (AR) pattern of inheritance. Consanguinity was noted in 63.9% of the families diagnosed with RP. The AR pattern of inheritance was observed in 96.1% of the RP families [14]. Consanguinity was observed in 11 of the 18 families with cone dystrophy [14].

Recurrent mutations characterized in genetic studies can be leveraged to develop screening tests for indigenous African populations. The mutations can be used together as targets in population screening. For example, the three recurrent mutations identified by Maggi et al. [19] combined account for 64% (9/14) of ORF15-associated indigenous African families characterized by 2020.

Tests developed based on genetic data collected from mainly Caucasians can only be useful among indigenous Africans if the targeted mutations are also established to be pathogenic mutations among the indigenous Africans. Even though the study by Roberts et al. [23] found that their locally developed test was efficacious in diagnosing AAR, they specified that the findings were only applicable to Caucasians in South Africa as they were the main population represented in the study. They recommended the investigation of the seven *ABCA4* mutations among indigenous Black Africans to compare frequencies in them with those observed among Caucasians and thus determine the usefulness of the developed test among the Africans [23]. Leveraging the *ABCA4* known sequence variants to facilitate screening of STGD among Africans requires genetic studies comprising mainly African probands to confirm that the *ABCA4* mutations targeted by the developed tests are also present in indigenous black Africans [24].

## 4. Discussion

The findings of this review show that most studies on genetics of IRDs in Africa have only been performed in South Africa and North Africa. Even for studies performed abroad but comprising some indigenous Africans, the indigenous African patients are mainly from North African countries [15]. In East and Central Africa, not a single study on the genetics of IRDs was retrieved. As an ophthalmologist practicing in Kenya and tackling IRDs, the primary author recognizes that the initial steps to embrace genetic testing to diagnose IRDs are being taken now in research projects he has initiated. Considering the importance of Africa’s genetic diversity in the development of genetic interventions including tests and treatments [25], it is vital to incorporate African genetic diversity in cohorts researching genetic diseases such as IRDs.

A laboratory dedicated to the study of the genetics of IRDs exists in South Africa. However, the main beneficiaries are Caucasians [23,24]. Indigenous Africans are minorities in genetic studies of IRDs in South Africa. According to Roberts et al. [3] and Greenberg et al. [3], most of the work carried out regarding the genetics of IRDs is among European-derived cohorts. There is a need to establish research centers on the genetics of IRDs in other parts of Africa such as Kenya to include more indigenous Africans in the studies. The data collected from diverse Africans and variants that will be discovered will contribute to overcoming the limitation of a few target genes and support the development of novel genetic tests and treatments for IRDs considering that Africans have more varied genomes [3,25,26].

### 4.1. Genes Affected in the Most Prevalent IRDs

In retinitis pigmentosa, the implicated genes that appeared across multiple studies include *MERTK*,* PDE6B*,* CERKL*, and *ABCA4* [14,15]. *MERTK* is important in molecular genetics diagnostics because bi-allelic pathogenic variants in *MERTK* have been linked to RP [27]. Missense and nonsense variants can occur in *MERTK* [28]. Multiple causative variants can occur in *MERTK*, some resulting from deletions [27]. Copy number variations (CNVs) in *MERTK* should be included in genetics studies of RP among Africans to capture the diverse mutations that can cause the disease [27]. The *PDE6B* gene can undergo missense and putative slicing defect mutations to cause RP [29]. High rates of occurrence of novel mutations in the *PDE6B* gene call for focus on it when studying genetic diversity in the incidence of RDs among Africans. Mutations in the *CERKL* gene are also substantially involved in the incidence of RP. A substantial proportion of mutations affecting the *CERKL* gene leading to RP are novel [30], which implies that there is a high likelihood of detecting multiple novel variants associated with the gene. Mutations in the *ABCA4* gene are implicated in Stargardt disease (STGD1), RP, and several other retinal dystrophies. More than 1200 mutations of various types including nonsense, missense, insertions, and splicing have been detected in the gene; they cause diverse severities of different diseases [31].

The genes that repeatedly appeared in the causation of LCA among Africans in the reviewed studies include *RPE65* and *GUCY2D* (LCA1) [18,21]. The first disease-causing genetic mutation was recognized in the *GUCY2D* gene [32]. Since then, several mutations occurring in more than 28 gene loci have been implicated in the causation of LCA [33]. *GUCY2D* and *RPE65* are among the commonly affected genes in LCA. Others include *CRB1, RDH12*, and *CEP290* [32]. However, the data on frequency is among Caucasians, hence the need to increase related research activities among indigenous Africans to identify the genes that frequently undergo mutations when LCA is developing.

The gene that emerged as involved in the incidence of Stargardt disease is *ABCA4*. The *C1490Y* is a recessive pathogenic variant in *ABCA4* that is associated with Stargardt macular dystrophy [34]. The *ABCA4* gene that codes for subfamily A, ATP-binding cassette, and member 4 transporter system undergo biallelic mutations resulting in pathogenicity in the retina that manifests as Stargardt disease [35]. The mutations exist in both coding and non-coding sequences of the *ABCA4* gene. They lead to multiple pathogenic variants in the gene’s intronic sequences [36]. The variants in the *ABCA4* gene cause aberrant splicing that results in defects characterizing Stargardt disease [35].

In code dystrophy among Africans, the genes involved in the pathogenesis include *ABCA4* and *KCNV12* [14,15]. Mutations in about 30 genes cause cone and cone-rod dystrophies [37]. Mutations in the *ABCA4* gene are associated with early onset and fast-progressing cone-rod dystrophy [31]. They account for about two-thirds of the autosomal recessive cone dystrophies and cone-rod dystrophies [37]. *KCNV2* is involved in the causation of cone dystrophy with supernormal rod response [31]. The *KCNV2*-associated retinopathy is an unusual cone-rod dystrophy characterized by supernormal rod responses and nyctalopia [38]. Severe cone-rod dystrophy may be difficult to distinguish from RP, a rod-cone dystrophy [31].

### 4.2. Choice of Test

Next generation sequencing and whole exome sequencing are the main methods of genetically diagnosing IRDs [14]. Sanger sequencing is used to confirm the presence of disease-causing variants detected by NGS or WES as it has a very high per-base accuracy [39]. NGS is more accessible and cheaper than Sanger sequencing for high numbers of targets, hence it is regularly used in clinical diagnosis [39]. WES is more sensitive and specific in the diagnosis of IRDs because it covers all protein-coding genes of the human genome, hence it is preferred as the primary test for IRDs [40].

The number of genes covered in NGS panels used for diagnosis of IRDs is increasing as the tests become cheaper [40,41]. However, the over 400 genes in the current NGS panels for IRDs are mainly obtained from research among Caucasians [41], hence the need for research among indigenous Africans to add genes specific to them in the panels. As African ophthalmologists are advancing in the diagnosis of IRDs using OCTs, OCTAs, and ERGs, they need to strengthen the accuracy and specificity of their diagnoses by incorporating genetic testing results based on genes characterized among indigenous Africans [14,42].

Mutation-targeting rapid genetic tests are also increasingly emerging [16]. If a unique variant is suspected as the underlying cause of an IRD, restriction fragment length polymorphism (RFLP) techniques and Sanger sequencing can be applied for single variant analysis in rapid and cost-effective genetic testing [39]. After a diagnosis has been identified through WES [39], single variant analysis can be applied to confirm the diagnosis. 

In cases where the standard pipeline of NGS is not accurate in detecting pathogenic mutations because of factors such as amplification failure, a novel pipeline such as the one used by Maggi [19] can be applied. In their study, the novel pipeline had 177 true positive variant calls compared with the standard pipeline’s 86. The novel pipeline only had 67 false positive variant calls, yet the standard pipeline had 520 [19].

### 4.3. Uniqueness of IRD-Causing Mutations among Africans 

Some mutations detected among Africans could be present in other populations, while others may be uniquely present among indigenous Africans. The heterozygous *CRB1* mutation c.1690G > T (p.Asp564Tyr) in one of the three Moroccan families with RP was first described in Spain. Perhaps the families were related, but some had migrated between Morocco and Spain [16]. On the other hand, the case with the *PDE6B* c.1920+2T > C mutation from the Moroccan families was the first of its kind in North Africa [16].

The *ABCA4* compound mutations c.5908C > T (p.Leu1970Phe) and c.6148G > C (p.Val2050Leu) identified in a case among the three Moroccan families have been individually characterized in other patients, but existence as a combination was unique [16]. The individual mutations may not be pathogenic because the parents of the case were not affected phenotypically [16]. The findings reveal the genetic diversity among Africans and the need to genetically characterize IRDs among Africans to enhance genetic diagnosis and treatment.

The need to focus on studying the genetics of IRDs among indigenous Africans manifests in the heterogeneity observed when Africans were included in previous genetic studies. Camuzat et al. [21] observed that, while the *LCA1* gene could be linked to LCA among North African patients, it could be excluded in all European patients except one. 

The concentration of IRD genetic studies in North Africa and South Africa could be because of the availability of laboratories that focus on oculogenetics in the countries. For example, the Hedi Rais Institute of Ophthalmology in Tunisia has an oculogenetics laboratory where patients with IRDs seek consultation. Leveraging the laboratory, Falfoul et al. [22] reported the genetic characteristics of the first cohort of patients phenotypically diagnosed with enhanced S-cone syndrome (ESCS) in Tunisia. Similarly, the Division of Human Genetics in the University of Cape Town in South Africa has a retinal research group and a laboratory that continues with the work of identifying specific mutations causing IRDs in South Africa [43], which was started about three decades ago. Therefore, the establishment of oculogenetics research groups laboratories in other parts of Africa such as East Africa, Central Africa, and West Africa could be instrumental in the understanding, diagnosis, and treatment of IRDs in Africa.

## 5. Conclusions

In conclusion, IRDs are an unresolved problem affecting the world including Africa. They are reducing the quality of life of the affected people, hence the need for timely and effective interventions to address the problem. Advances in gene therapy are benefiting IRD patients in countries where research on IRDs is advanced. Identification of the most appropriate gene therapy requires genetic diagnoses of IRD patients. Thus, genetic testing results are a prerequisite in participation in global clinical trials testing the effectiveness of gene therapy to treat IRDs. In Africa, genetic studies of IRDs are mainly performed in South Africa and North Africa. Thus, indigenous Africans are scantly represented in the cohorts of IRD patients from whom the data used to develop genetic tests and therapies for IRDs are collected. There is a need for genetic IRD research activities in other parts of Africa including East Africa, Central Africa, and West Africa. The genetic research should specifically target indigenous black Africans. 

## Figures and Tables

**Figure 1 jpm-13-00239-f001:**
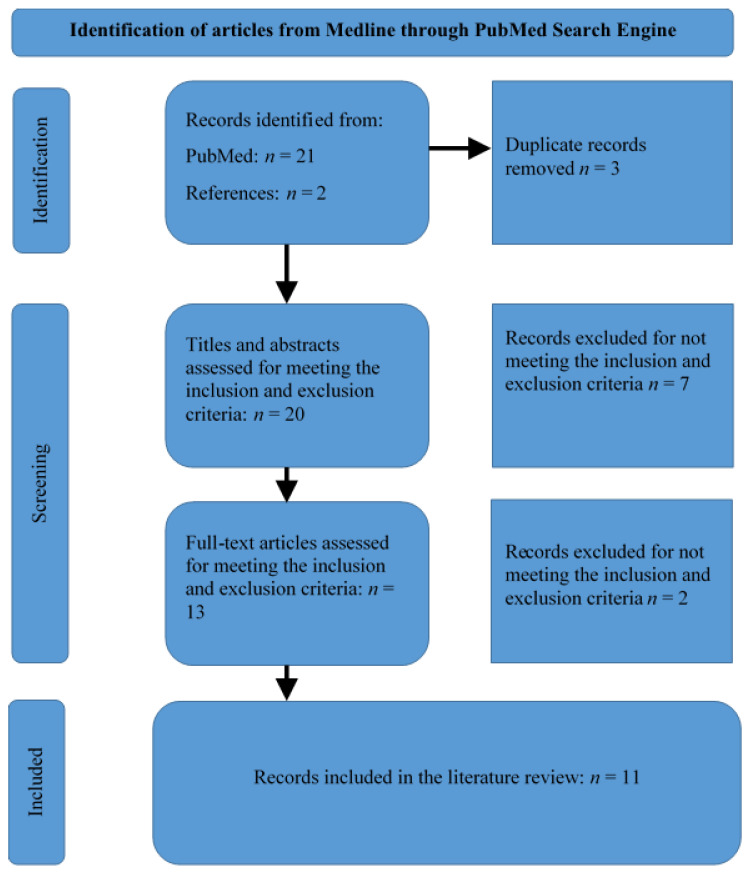
Selection of articles for the review.

**Figure 2 jpm-13-00239-f002:**
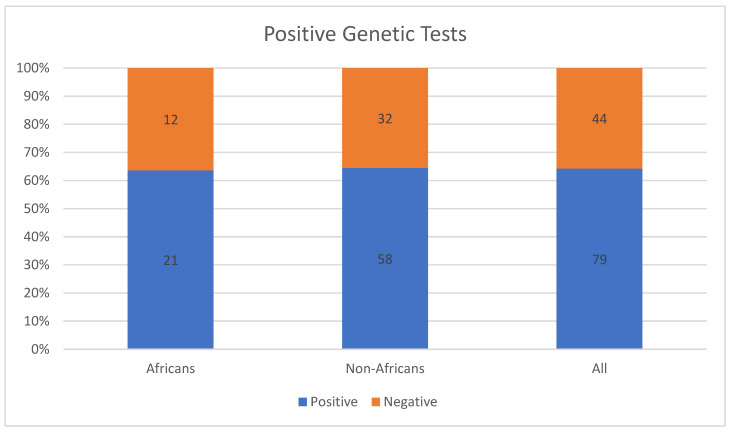
Proportions of positive genetic tests for IRDs among Africans and non-Africans in the study by Maltese et al. [15].

**Figure 3 jpm-13-00239-f003:**
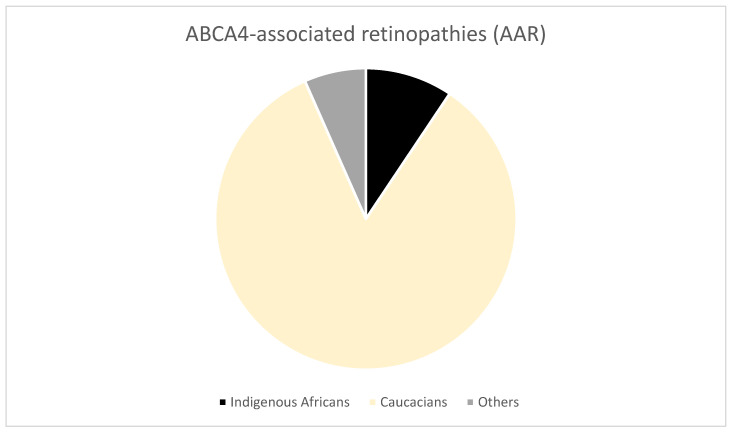
A pie chart showing the proportions of ethnicities of probands with AAR included in the study by Roberts et al. [23].

**Table 1 jpm-13-00239-t001:** Summary of studies addressing the topic of genetic testing of inherited retinal disorders (IRDs) among Africans.

Authors	Location of patients	Diseases	Included Africans	Genetic Testing Methods Used
Bouzidi et al. [14]	North Africa	IRDs and inherited optical neuropathies (ION)	413 families	Next-generation sequencing
Maltese et al. [15]	North Africa	IRDs	33 patients	Next-generation sequencing (NGS) and gene sequencing panels targeting a specific set of genes, Sanger sequencing and—when necessary—multiplex ligation-dependent probe amplification (MLPA)
Bouzidi et al. [16]	Morocco	Retinitis pigmentosa	3 families	Whole exome sequencing
Greenberg et al. [17]	South Africa	Retinitis pigmentosa	75 families	Unspecified DNA banking methods
Perrault et al. [18]	North Africa	Leber congenital amaurosis	7 families	Sequencing using primers specific to the cDNA sequence and mutation screening
Maggi et al. [19]	South Africa	Retinitis pigmentosa and recessive macular degeneration	RP: 17 patients MD: 15 probands	Next-generation PCR and sequencing
McKie et al. [20]	South Africa	Retinitis pigmentosa	1 family	PCR, segregation analysis, reverse transcriptase PCR, and Nothern blot analysis
Camuzat et al. [21]	North Africa	Leber congenital amaurosis	7 families	Genotyping using PCR applying hypervariable microsatellites
Falfoul et al. [22]	Tunisia	Enhance-S-cone syndrome	6 families	Unspecified genetic tests
Roberts et al. [23]	South Africa	*ABCA4*-associated retinopathies	17 probands	Rapid test to detect seven *ABCA4* mutations
September et al. [24]	South Africa	Stargardt disease	0 probands	Single-strand conformational polymorphism–heteroduplex analysis sequencing and restriction fragment length polymorphism analysis

## Data Availability

No new data were created.

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
