# Peer review of "Challenges and Opportunities in the Genetic Analysis of Inherited Retinal Dystrophies in Africa, a Literature Review"

_jpm, 2023, doi:10.3390/jpm13020239_

Round 1
Reviewer 1 Report
The manuscript “ Challenges and Opportunities in the Genetic Analysis of Inherited Retinal Dystrophies in Africa, A Literature Review” introduced the genetic researches on IRDs, especially in East, Central, and West Africa.
It is a useful topic. This review aims to synthesize the findings of empirical studies that have been done through genetic testing to characterize inherited retinal dystrophies in Africa. It will reveal knowledge gaps that ophthalmology researchers can focus on filling through exploratory and experimental research on the genetics of IRDs. However, there are some problems need to be address:
1. The content of the “Introduction section” can be stated more briefly to highlight the important description.
2. The description of the “Results section” can be presented in table form to make the description clearer.
3. There are relatively few references that can be cited in this review, and more relevant studies are needed to illustrate the conclusions of this review.
4. Are “Discussion sections” 4.1-4.3 specific to the disease in Africa or globally? It is recommended that some paragraphs be clearly stated.
Reviewer 2 Report
This review paper provides a well-written and comprehensive examination of the current state of genetic research on inherited retinal dystrophies in Africa. The authors have done an excellent job of synthesizing the existing amount of literature on the topic and highlighting key findings.
The article effectively describes the genetic diversity and complexity of inherited retinal dystrophies in Africa and the challenges that this presents for genetic analysis. The authors also provide a thorough overview of current genetic testing methods and their limitations in the African population, as well as the underrepresentation of African indigenous population even in studies performed in African centers.
One of the strengths of this review is the emphasis on the importance of collaboration and capacity building between African and international researchers in order to address the challenges and opportunities in this field. This is an important message that highlights the need for continued investment in genetic research in Africa.
Overall, this review provides a good overview of the current state of genetic research on inherited retinal dystrophies in Africa and offers valuable insights for researchers and clinicians working in this field.
General comments:
It would be useful to provide the reader with some figures summarizing the main findings of this review (e.g. pie charts representing the proportions of the most common mutations involved in IRDs in Africa vs Europe vs Asia).
Minor comments:
Line 348à Mutations in ABCA4 are implicated mainly in STGD1, not in RP
Line 380 à NGS is not cheaper and accessible than Sanger sequencing. It is actually the opposite.
